# Hepatoprotective Effects of *Glycyrrhiza glabra* in Diabetic Male Rats: Addressing Liver Function, Oxidative Stress, and Histopathological Changes

**DOI:** 10.3390/biology14030307

**Published:** 2025-03-18

**Authors:** Abdulmajeed F. Alrefaei, Mohamed E. Elbeeh

**Affiliations:** 1Department of Biology/Genetic and Molecular Biology Central Laboratory (GMCL), Jamoum University College, Umm Al-Qura University, Makkah 2203, Saudi Arabia; afrefaei@uqu.edu.sa; 2Department of Biology, Jamoum University College, Umm Al-Qura University, Makkah 21955, Saudi Arabia; 3Department of Zoology, Faculty of Science, Mansoura University, Mansoura 35516, Egypt

**Keywords:** *Glycyrrhiza glabra*, diabetes mellitus, hepatoprotection, oxidative stress, histopathology

## Abstract

This study demonstrated that *Glycyrrhiza glabra* (licorice) has strong hepatoprotective effects in diabetic male rats, significantly improving liver function by reducing enzyme levels (ALT, AST, ALP) and oxidative stress markers (MDA) while enhancing antioxidant activity (GSH, SOD, CAT). A histological analysis showed reduced liver inflammation, fat accumulation, and fibrosis in licorice-treated diabetic rats. Additionally, licorice improved glucose metabolism, insulin sensitivity, and lipid profiles. These findings highlight its potential as a natural therapeutic agent for managing diabetes-related liver complications, though further research is needed to confirm its long-term efficacy and safety in humans.

## 1. Introduction

Diabetes mellitus (DM) is a chronic, multifactorial metabolic disorder characterized by persistent hyperglycemia, insulin resistance, and the dysregulation of carbohydrate, lipid, and protein metabolism. This condition has far-reaching systemic effects, with the liver being particularly vulnerable due to its central role in glucose homeostasis, lipid metabolism, and detoxification processes [1,2]. Over time, chronic hyperglycemia and insulin resistance contribute to a spectrum of hepatic complications, including non-alcoholic fatty liver disease (NAFLD), hepatosteatosis, hepatic fibrosis, cirrhosis, and hepatocellular carcinoma (HCC) [3,4]. These complications significantly increase the risk of liver dysfunction, further worsening the metabolic and inflammatory burden in diabetic individuals, ultimately leading to end-stage liver disease if left untreated [5].

The pathogenesis of diabetes-associated liver disease is complex and involves a vicious cycle of oxidative stress, mitochondrial dysfunction, chronic inflammation, and fibrotic remodeling [6,7]. Oxidative stress plays a central role in the progression of hepatic damage, as the excessive production of reactive oxygen species (ROS) leads to lipid peroxidation, DNA fragmentation, and hepatocyte apoptosis [8]. Moreover, ROS accumulation activates inflammatory pathways, including nuclear factor-kappa B (NF-κB) and tumor necrosis factor-alpha (TNF-α), further exacerbating hepatocyte injury and fibrosis [9]. Insulin resistance, another major driver of hepatic complications, promotes hepatic steatosis by increasing lipogenesis and impairing fatty acid oxidation, creating a metabolic imbalance that accelerates lipotoxicity and hepatic dysfunction [10].

Researchers are exploring natural bioactive compounds for diabetes-related liver diseases due to their antioxidant, anti-inflammatory, and metabolic benefits [11]. Among these, *Glycyrrhiza glabra* (licorice) has emerged as a potent candidate, widely recognized in traditional and modern medicine for its diverse pharmacological properties, particularly in the treatment of hepatic and metabolic disorders [12].

The hepatoprotective effects of licorice are primarily attributed to its rich composition of bioactive phytochemicals, including glycyrrhizin, glabridin, and 18β-glycyrrhetinic acid, which exhibit strong antioxidant activity, free radical scavenging capacity, and the inhibition of lipid peroxidation [13,14]. These compounds enhance the endogenous antioxidant defense system by upregulating superoxide dismutase (SOD), catalase (CAT), and glutathione peroxidase (GPx), thereby reducing oxidative stress-induced hepatic damage [15]. Furthermore, licorice exerts profound anti-inflammatory effects by inhibiting the expression of pro-inflammatory cytokines such as interleukin-6 (IL-6), TNF-α, and NF-κB, which play a crucial role in fibrotic and inflammatory liver injury [16].

Beyond its hepatoprotective properties, *Glycyrrhiza glabra* has been extensively studied for its antidiabetic, lipid-lowering, neuroprotective, and anticancer effects, further supporting its potential as a broad-spectrum therapeutic agent [17]. Preclinical studies have demonstrated that licorice supplementation significantly reduces key markers of liver dysfunction, including serum alanine aminotransferase (ALT), aspartate aminotransferase (AST), and alkaline phosphatase (ALP), while restoring hepatic lipid balance, improving glucose metabolism, and reducing hepatic triglyceride accumulation [18]. Additionally, glycyrrhizin and glabridin have been shown to regulate lipid metabolism by inhibiting lipogenesis, enhancing mitochondrial function, and preventing hepatic steatosis, reinforcing their role in protecting the liver from diabetes-induced damage [19]. Oxidative stress and inflammation are closely interconnected processes that play a pivotal role in the progression of liver diseases associated with diabetes mellitus. The excessive production of reactive oxygen species (ROS) leads to lipid peroxidation, DNA damage, and hepatocyte apoptosis, which subsequently activate inflammatory pathways, including the nuclear factor-kappa B (NF-κB) signaling cascade. This inflammatory response further exacerbates hepatic injury, creating a vicious cycle that accelerates liver dysfunction. Given the significant role of oxidative stress and inflammation in hepatic complications, the exploration of natural antioxidants with strong free radical scavenging and anti-inflammatory properties has gained considerable attention. *Glycyrrhiza glabra* (licorice), known for its rich composition of bioactive phytochemicals, represents a promising natural candidate for interrupting this pathological cycle by reducing oxidative damage and modulating inflammatory responses [15].

Despite these promising findings, additional research is needed to fully elucidate the molecular mechanisms underlying the hepatoprotective effects of *Glycyrrhiza glabra*, as well as to determine its optimal dosage, pharmacokinetics, and long-term safety profile in clinical applications.

This study aimed to comprehensively evaluate the hepatoprotective effects of *Glycyrrhiza glabra* in an experimental model of diabetes-induced liver injury, focusing on its impact on oxidative stress, hepatic inflammation, fibrotic progression, and metabolic dysfunction. By bridging the gap between traditional medicinal knowledge and modern pharmacological research, this study seeks to provide scientific validation for the therapeutic use of licorice in managing diabetes-related hepatic complications. The findings may pave the way for the development of plant-based interventions that offer safe, effective, and accessible alternatives to conventional pharmacotherapy in treating diabetes-induced liver disorders.

## 2. Methods

### 2.1. Study Design and Experimental Setup

This study was conducted using forty male albino Sprague Dawley rats, each weighing between 180 and 200 g and aged six months. The animals were obtained from the Animal Health Research Institute and housed in a controlled environment with a temperature range of 22–25 °C, relative humidity of 50–60%, and a 12 h light–dark cycle. They were maintained on a standard commercial diet and had ad libitum access to water throughout the experiment to ensure adequate nutritional status and minimize external metabolic influences. Ethical approval was obtained from the Animal Ethics Committee of Umm Al-Qura University (Approval No. HAPO-02-K-010-2023-02-1657), ensuring compliance with institutional and international animal care guidelines for humane handling and experimental procedures [20].

Diabetes was induced using a single intraperitoneal injection of streptozotocin (STZ) at a dose of 60 mg/kg after a 12 h fasting period. STZ, a compound known for its ability to selectively target pancreatic β-cells through oxidative stress mechanisms, effectively creates a reproducible model of Type 1 diabetes mellitus in rodents. Following STZ administration, fasting blood glucose (FBG) levels were measured, and rats with FBG exceeding 240 mg/dL were confirmed as diabetic and included in the experimental groups [21].

The aqueous extract of *Glycyrrhiza glabra* was prepared through a standardized protocol. Licorice roots were thoroughly washed with distilled water to remove impurities and air-dried in shade at room temperature (25 °C) for seven days to preserve bioactive compounds. The dried roots were then ground into a fine powder using an electric grinder. Extraction was performed using distilled water at a ratio of 1:10 (*w*/*v*) at 60 °C for two hours under continuous stirring. The extract was filtered through Whatman No. 1 filter paper, concentrated at 50 °C under reduced pressure using a rotary evaporator, and subsequently freeze-dried to obtain a stable powdered extract. A stock solution of 100 mg/mL was prepared and diluted to the experimental dose of 250 mg/kg/day. The presence of glycyrrhizin and other active constituents in the extract was confirmed using a high-performance liquid chromatography (HPLC) analysis [22].

Rats were divided into four experimental groups, with ten rats in each category: a control group (C) consisting of non-diabetic untreated rats, a *Glycyrrhiza glabra*-treated group (G) comprising non-diabetic rats receiving 250 mg/kg/day of the extract, a diabetic untreated group (D), and a diabetic group treated with 250 mg/kg/day of *Glycyrrhiza glabra* extract (DG). The extract was administered orally via a gastric tube for eight weeks, beginning two weeks post-diabetes induction. At the conclusion of the study, rats were euthanized under anesthesia, and blood samples were collected via cardiac puncture for biochemical analysis. Liver tissues were immediately fixed in 10% neutral buffered formalin for 24–48 h to ensure proper tissue preservation for histopathological examination. Post-fixation, samples were washed with distilled water, dehydrated using a graded ethanol series (70–100%), cleared with xylene, and embedded in paraffin wax. Thin sections (5 µm) were prepared using a Leica RM2235 microtome, stained with hematoxylin and eosin (H&E), and examined under an Olympus CX43 microscope to evaluate histopathological changes such as inflammation, fatty degeneration, and fibrosis [23].

### 2.2. RNA Extraction and cDNA Synthesis

Total RNA was extracted from liver tissues using TRIzol™ reagent (Invitrogen, Thermo Fisher Scientific, Waltham, MA, USA) according to the manufacturer’s protocol. RNA concentration and purity were determined using a NanoDrop spectrophotometer (Thermo Fisher Scientific, USA). One microgram of RNA was reverse transcribed into complementary DNA (cDNA) using the High-Capacity cDNA Reverse Transcription Kit (Applied Biosystems, Thermo Fisher Scientific, USA). The cDNA samples were stored at −20 °C until further use for gene expression analysis [24].

### 2.3. Quantitative Real-Time PCR (qRT-PCR)

Gene expression levels related to liver function and oxidative stress were evaluated using quantitative real-time PCR (qRT-PCR). Amplification reactions were carried out using an Applied Biosystems QuantStudio™ 3 instrument with PowerUp™ SYBR™ Green Master Mix (Thermo Fisher Scientific, USA). Primer sequences were designed based on publicly available gene sequences from the NCBI database and were commercially synthesized by Integrated DNA Technologies (IDT, Coralville, Iowa, USA). The thermal cycling protocol included an initial denaturation at 95 °C for 3 min, followed by 40 cycles of denaturation at 95 °C for 15 s, annealing at specific primer temperatures, and extension at 72 °C for 30 s. Specificity was confirmed through melting curve analysis, and gene expression was normalized to β-actin using the ΔΔCt method [25].

### 2.4. Biochemical Analyses

Blood glucose levels were monitored weekly using an Accu-Chek Active glucometer (Roche Diagnostics, Rotkreuz, Switzerland) following standard protocols. Serum biochemical markers of liver function, including alanine aminotransferase (ALT), aspartate aminotransferase (AST), alkaline phosphatase (ALP), and bilirubin, were quantified using commercially available assay kits from BioAssay Systems, USA. Oxidative stress markers, such as malondialdehyde (MDA), and antioxidant enzyme activities, including superoxide dismutase (SOD) and catalase (CAT), were measured using TBARS Assay Kits from Cayman Chemical, USA, and Sigma-Aldrich Kits (S5639 for SOD and CAT100 for CAT) [26].

### 2.5. Histological and Comet Assay Analysis

A histological analysis of liver tissue was conducted to assess pathological changes associated with diabetes and *Glycyrrhiza glabra* treatment. Liver sections were stained with H&E and examined under light microscopy for signs of necrosis, inflammatory cell infiltration, and fibrosis. The comet assay, a gel electrophoresis-based technique, was performed to evaluate DNA damage in individual liver cells. Liver tissue was homogenized to prepare single-cell suspensions, which were embedded in low-melting-point agarose on microscope slides. Cells were lysed and subjected to electrophoresis under alkaline conditions, followed by neutralization and staining with SYBR Green (Invitrogen, USA). Fluorescence microscopy was used to capture images, and DNA damage was quantified using Comet Assay IV software (Instem, Instem, Stone, Staffordshire, UK, version 4.3). The comet assay was performed within 24 h to ensure accurate assessment of DNA strand breaks and oxidative damage [27].

### 2.6. Statistical Analysis

All statistical analyses were conducted using GraphPad Prism 9.0 software (GraphPad Software Inc., San Diego, CA, USA). Data were expressed as mean ± standard deviation (SD), and intergroup differences were analyzed using one-way analysis of variance (ANOVA), followed by Tukey’s post hoc test for multiple comparisons. A *p*-value of <0.05 was considered statistically significant. Statistical interpretations were made based on guidelines outlined by Motulsky for the appropriate analysis of biological data [28].

## 3. Results

### 3.1. Effect of Glycyrrhiza glabra on Metabolic and Physical Parameters

The body weight of the rats was recorded throughout the experiment, as shown in Figure 1A. After diabetes induction with streptozotocin (STZ), diabetic (D) rats showed a rapid and significant reduction in body weight compared to the control (C) group (*p* < 0.01). Conversely, rats treated with *Glycyrrhiza glabra* (D+G group) exhibited a significant attenuation in weight loss compared to the untreated diabetic rats (*p* < 0.05). By the end of the experiment (week 12), the mean body weights were approximately 350 ± 10 g for the control group, 340 ± 12 g for the *Glycyrrhiza*-treated group (G), 220 ± 20 g for the diabetic group (D), and 300 ± 15 g for the D+G group. These results demonstrate that *Glycyrrhiza glabra* significantly mitigates weight loss associated with diabetes.

Fasting blood glucose (FBG) levels were also monitored and are presented in Figure 1B. The diabetic (D) group exhibited a marked elevation in FBG levels compared to the control group throughout the study (*p* < 0.01). However, treatment with *Glycyrrhiza glabra* (D+G group) significantly reduced FBG levels compared to the untreated diabetic group (*p* < 0.05). On day 45, the mean FBG levels were approximately 90 ± 5 mmol/L for the control group, 85 ± 6 mmol/L for the G group, 240 ± 15 mmol/L for the D group, and 150 ± 10 mmol/L for the D+G group. These findings indicate the glycemic control potential of *Glycyrrhiza glabra* in diabetic rats.

### 3.2. Effects of Glycyrrhiza glabra on Glucose Tolerance and Insulin Sensitivity

The effects of *Glycyrrhiza glabra* on insulin sensitivity and β-cell function were assessed using HOMA-IR and HOMA-β indices. As shown in Figure 2A, rats in the *Glycyrrhiza glabra*-treated group (G group) exhibited a significant reduction in HOMA-IR levels compared to the diabetic group (D group) (*p* < 0.0001), indicating improved insulin sensitivity. Similarly, HOMA-β values were significantly higher in the G group compared to the D group (*p* < 0.0001; Figure 2B), reflecting enhanced β-cell function.

Figure 2C–F present the results of the Oral Glucose Tolerance Test (OGTT) and Insulin Sensitivity Test (IST). The D group displayed a significantly elevated glycemic response to glucose administration compared to the control group (C group). However, *Glycyrrhiza glabra*-treated rats (D+G group) demonstrated significantly reduced glycemic responses across the 120 min observation period during the OGTT (*p* < 0.01; Figure 2E). Additionally, the D+G group showed a significant improvement in the Insulin Sensitivity Test (IST) results compared to the untreated D group (*p* < 0.05; Figure 2F).

The area under the curve (AUC) calculations for OGTT and IST, as shown in Figure 2C,D, further corroborate these findings, with significant improvements observed in the D+G group compared to the D group. These results suggest that *Glycyrrhiza glabra* effectively ameliorates glucose metabolism disturbances, improving insulin sensitivity and reducing glycemic response, which are critical in minimizing risk factors associated with diabetes-induced liver damage.

### 3.3. Impact of Glycyrrhiza glabra on Serum Lipids in Diabetic Rats

Dysregulated lipid metabolism is a hallmark of diabetes, as evidenced by the serum lipid profile in diabetic rats (D group). As shown in Figure 3A–E, diabetic rats exhibited significantly elevated levels of total cholesterol (TC), triglycerides (TG), free fatty acids (FFAs), and low-density lipoprotein (LDL) compared to the control group (C group) (*p* < 0.01). In contrast, their levels of high-density lipoprotein (HDL) were significantly reduced (*p* < 0.01), confirming the presence of substantial dyslipidemia induced by the combination of a high-fat diet (HFD) and streptozotocin (STZ) treatment.

Treatment with an oral water extract of *Glycyrrhiza glabra* (G group) for six weeks significantly improved the lipid profile of diabetic rats. Compared to the D group, rats in the G group showed marked reductions in TG, TC, LDL, and FFA levels, along with a significant increase in HDL levels (*p* < 0.05). Specifically, *Glycyrrhiza glabra* extract mitigated the levels of TG, TC, LDL, HDL, and FFA by 18.55%, 24.04%, 23.39%, 25.13%, and 31.54%, respectively. These results suggest that *Glycyrrhiza glabra* effectively ameliorates dyslipidemia associated with diabetes and enhances lipid homeostasis.

### 3.4. Impact on Antioxidant Enzymes in Diabetic Rats

The effects of *Glycyrrhiza glabra* on antioxidant enzymes in diabetic rats are presented in Figure 4A–D. Diabetic rats (D group) exhibited significantly reduced activities of catalase (CAT), superoxide dismutase (SOD), and glutathione peroxidase (GPx) compared to the control group (C group) (*p* < 0.01), indicating impaired antioxidant defenses.

Treatment with an oral water extract of *Glycyrrhiza glabra* (G group) significantly enhanced the activities of SOD, CAT, and GPx in diabetic rats compared to the untreated diabetic group (D group) (*p* < 0.05). This improvement in antioxidant enzyme activities suggests an enhanced ability to combat oxidative stress in diabetic rats treated with *Glycyrrhiza glabra*.

Interestingly, the activity of Glutathione-S-Transferase (GST) was significantly elevated in the D group compared to both the control (C) and *Glycyrrhiza glabra*-treated (G) groups (*p* < 0.05). These results highlight the antioxidant potential of *Glycyrrhiza glabra* in alleviating oxidative stress and restoring antioxidant defenses in diabetic rats.

### 3.5. The Impact of Glycyrrhiza glabra on Gene Expression: A Comprehensive Analysis

This study demonstrates the strong hepatoprotective effects of *Glycyrrhiza glabra* in a diabetic rat model, highlighting its role in improving liver function, lipid metabolism, and oxidative stress. Notably, rats treated with *Glycyrrhiza glabra* exhibited significant upregulation of key antioxidant defense genes, such as SOD1 and CAT, indicating enhanced antioxidant capacity and reduced oxidative damage.

Moreover, the licorice-treated group displayed significantly elevated expression of IGF1 and GHr, which are critical markers of liver growth and regeneration. These findings suggest the restoration of growth-related pathways and the potential of *Glycyrrhiza glabra* to promote liver repair in diabetes-induced damage.

Improved metabolic regulation was evident in the increased expression of PPARα and LPL, which are associated with enhanced lipid metabolism, and the downregulation of FAS, indicating reduced lipogenesis. These results underscore the role of *Glycyrrhiza glabra* in restoring lipid homeostasis and mitigating dyslipidemia in diabetic conditions.

Collectively, these findings demonstrate the therapeutic potential of *Glycyrrhiza glabra*. By enhancing metabolism, reducing free radical damage, and promoting cellular repair, *Glycyrrhiza glabra* facilitates the recovery of liver function and protects against diabetes-induced liver damage (Figure 5).

### 3.6. Assessment of Liver Health Parameters

The analysis of liver health markers in diabetic rats treated with *Glycyrrhiza glabra* (DG group) revealed significant improvements compared to the untreated diabetic group (D group). Serum levels of liver enzymes, including alanine aminotransferase (ALT), aspartate aminotransferase (AST), and alkaline phosphatase (ALP), were significantly reduced in the DG group (*p* < 0.05), reflecting enhanced liver function.

Oxidative stress markers showed a similar trend of improvement. Malondialdehyde (MDA), a key indicator of lipid peroxidation and oxidative stress, was significantly lower in the DG group compared to the D group (*p* < 0.01). Meanwhile, glutathione (GSH), a critical antioxidant defense molecule, was significantly elevated in the DG group (*p* < 0.05). These findings demonstrate the capacity of *Glycyrrhiza glabra* to mitigate oxidative damage by enhancing antioxidant defenses and reducing lipid peroxidation in diabetic liver tissues.

A histopathological examination further highlighted the hepatoprotective effects of *Glycyrrhiza glabra* (Figure 6A–G). Liver sections from diabetic rats (D group) exhibited severe pathological alterations, including excessive fat accumulation, inflammation, and fibrosis. However, treatment with *Glycyrrhiza glabra* significantly alleviated these changes, restoring normal liver architecture, reducing fat deposition, and suppressing inflammatory infiltration and fibrosis.

These results collectively underscore the therapeutic potential of *Glycyrrhiza glabra* in protecting against diabetes-induced liver damage. By scavenging free radicals, reducing oxidative stress, and improving liver structure and function, *Glycyrrhiza glabra* offers a promising approach for mitigating liver complications associated with diabetes.

### 3.7. Comet Assay of Liver Tissue in Diabetic Rats

The comet assay revealed substantial DNA damage in the liver cells of diabetic rats (D group), characterized by significantly longer apoptotic tail lengths compared to the control group (C group). This indicates severe DNA strand breaks and oxidative damage induced by diabetes. Conversely, treatment with *Glycyrrhiza glabra* significantly reduced DNA damage in the D+G group (*p* < 0.05), as evidenced by shorter apoptotic tails. These findings support the hepatoprotective role of *Glycyrrhiza glabra* in mitigating DNA damage caused by diabetes.

### 3.8. Apoptotic Tail Length and Radiation Response

The apoptotic tail length was analyzed in liver cells from the D and D+G groups after exposure to varying doses of radiation (2, 8, 30, and 60 Gy). Figure 7A presents dose–response curves that illustrate the relationship between radiation dose and apoptotic tail length for each group. The results demonstrate a clear reduction in tail length in the D+G group compared to the D group, suggesting that *Glycyrrhiza glabra* provides significant protection against DNA damage induced by both diabetes and radiation.

### 3.9. DNA Damage Assessment Methodology

The comet assay was performed using the alkaline overnight method, which detects DNA single-strand breaks, double-strand breaks, and alkali-labile lesions. Single-cell suspensions were exposed to X-rays on ice to prevent DNA strand rejoining, allowing for accurate measurement of damage. Apoptotic tail lengths were then quantified for each group.

These findings confirm the ability of *Glycyrrhiza glabra* to mitigate DNA damage in diabetic rats, emphasizing its potential as a protective agent against both diabetes- and radiation-induced oxidative stress.

## 4. Discussion

This study provides robust evidence for the hepatoprotective effects of *Glycyrrhiza glabra* in a diabetic rat model, demonstrating its ability to restore liver function, mitigate oxidative stress, and preserve histological integrity. The results support its therapeutic potential in managing diabetes-induced hepatic complications, with glycyrrhizin identified as the primary active compound responsible for these effects.

### 4.1. Mechanistic Insights

The mechanisms through which *Glycyrrhiza glabra* exerts its benefits include antioxidant activity, anti-inflammatory effects, insulin sensitization, and lipid metabolism regulation. These findings are consistent with previous research indicating that *Glycyrrhiza glabra* enhances liver function, stabilizes hepatocyte membranes, and reduces inflammation through its bioactive components [29,30].

### 4.2. Restoration of Liver Function

Diabetes mellitus induces hepatotoxicity, which is evident from elevated serum levels of alanine aminotransferase (ALT), aspartate aminotransferase (AST), and alkaline phosphatase (ALP). These enzymes serve as biomarkers of hepatic injury, with increased levels indicating hepatocellular damage and impaired liver function. Treatment with *Glycyrrhiza glabra* significantly lowered ALT, AST, and ALP levels, suggesting that the extract protects hepatocytes from oxidative and inflammatory damage. This aligns with prior studies reporting that glycyrrhizin stabilizes hepatocyte membranes, reduces hepatic inflammation, and enhances enzymatic regulation [31].

### 4.3. Reduction in Oxidative Stress

A key pathological feature of diabetes-associated liver dysfunction is oxidative stress, characterized by an imbalance between reactive oxygen species (ROS) production and antioxidant defenses. Elevated malondialdehyde (MDA) levels, a marker of lipid peroxidation, indicate heightened oxidative damage. In this study, *Glycyrrhiza glabra* administration significantly reduced MDA levels, reflecting a reduction in lipid peroxidation and oxidative injury. Concurrently, glutathione (GSH) and catalase (CAT) levels were significantly increased, indicating enhanced endogenous antioxidant defenses. These findings corroborate previous studies showing that glycyrrhizin and other flavonoids in *Glycyrrhiza glabra* upregulate antioxidant enzymes, neutralize ROS, and protect hepatocytes from oxidative injury [32,33].

### 4.4. Histopathological Improvements

A histological examination of liver tissues further validated the hepatoprotective properties of *Glycyrrhiza glabra*. Diabetic rats exhibited severe histopathological abnormalities, including fat accumulation (steatosis), hepatocyte necrosis, inflammatory infiltration, and fibrosis. However, *Glycyrrhiza glabra* treatment significantly improved liver morphology, demonstrating reduced fat deposition, alleviated inflammation, and decreased fibrosis. These results suggest that *Glycyrrhiza glabra* inhibits hepatic inflammation and promotes tissue repair. Prior research has shown that glycyrrhizin downregulates pro-inflammatory cytokines, such as tumor necrosis factor-alpha (TNF-α) and interleukin-6 (IL-6), while simultaneously inhibiting fibrotic pathways, including transforming growth factor-beta 1 (TGF-β1) signaling [34,35].

### 4.5. Proposed Mechanisms of Hepatoprotection

The hepatoprotective effects of *Glycyrrhiza glabra* can be attributed to its bioactive compounds, which exert their effects through multiple mechanisms:

Reduction in Oxidative Stress: *Glycyrrhiza glabra* enhances mitochondrial function, upregulates antioxidant enzymes (SOD, CAT, GPx), and prevents ROS-induced hepatocyte damage, thus mitigating oxidative stress [36].

Anti-Inflammatory Actions: The extract inhibits the activation of nuclear factor-kappa B (NF-κB), a key regulator of inflammation, leading to reduced TNF-α and IL-6 expression, thereby suppressing hepatic inflammation [37].

Improved Lipid Metabolism: The upregulation of peroxisome proliferator-activated receptor alpha (PPARα) and lipoprotein lipase (LPL) expression enhances fatty acid oxidation and lipid clearance, while reduced fatty acid synthase (FAS) expression decreases lipogenesis and steatosis [38].

Insulin Sensitization: *Glycyrrhiza glabra* modulates insulin signaling pathways, enhances glucose uptake, and reduces insulin resistance, thereby mitigating diabetes-induced hepatic dysfunction [39].

### 4.6. Comparison with Previous Studies

The current findings align with previous research highlighting the hepatoprotective effects of *Glycyrrhiza glabra* in diabetic models. A study by Yang et al. [40] reported significant reductions in ALT and AST levels following glycyrrhizin administration in diabetic rats, while Huo et al. [41] demonstrated that glycyrrhizin attenuates liver fibrosis by inhibiting TGF-β1 signaling. The present study expands upon these findings by incorporating molecular analyses of oxidative stress markers, lipid metabolism regulation, and inflammatory cytokine modulation, providing a more comprehensive understanding of *Glycyrrhiza glabra*’s hepatoprotective potential.

### 4.7. Limitations of the Study

Despite its promising results, this study has certain limitations. The use of a diabetic rat model may not fully replicate the complexity of human diabetes and its associated liver complications. Additionally, the study duration was relatively short, precluding an assessment of long-term hepatic outcomes. Furthermore, only one dose of *Glycyrrhiza glabra* was tested, leaving uncertainty regarding the optimal dosage and dose–response relationship.

### 4.8. Future Research Directions

Molecular Mechanisms: Further studies utilizing transcriptomics and proteomics could provide deeper insights into the molecular pathways modulated by *Glycyrrhiza glabra*.

Clinical Trials: Long-term human trials are essential to confirm its safety and efficacy in managing diabetes-related liver complications.

Combination Therapies: Future research should explore synergistic effects between *Glycyrrhiza glabra* and conventional antidiabetic or hepatoprotective agents.

Dose Optimization: Further studies should investigate the dose–response relationship and determine the optimal therapeutic dosage for maximizing efficacy while minimizing potential side effects.

## 5. Conclusions

This study demonstrates that *Glycyrrhiza glabra* exerts significant hepatoprotective effects in a diabetic rat model by enhancing liver function, reducing oxidative stress, and alleviating histopathological damage. The bioactive compounds of *Glycyrrhiza glabra*, particularly glycyrrhizin, contribute to its antioxidant, anti-inflammatory, insulin-sensitizing, and lipid-modulating properties. These findings provide scientific validation for the traditional use of *Glycyrrhiza glabra* in hepatoprotection and highlight its potential as a therapeutic agent for diabetes-associated liver disorders. However, further clinical research, mechanistic studies, and long-term evaluations are necessary to confirm its therapeutic application, establish optimal dosing regimens, and ensure safety in human populations.

## Figures and Tables

**Figure 1 biology-14-00307-f001:**
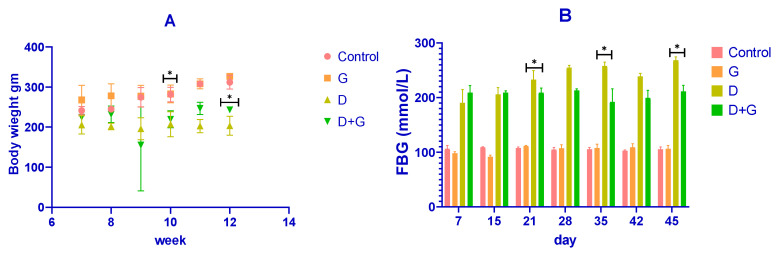
The figure demonstrates the effects of *Glycyrrhiza glabra* (G) on body weight and fasting blood glucose (FBG) levels in different rat groups. Subfigure (**A**): Displays the changes in body weight over the experimental period. Rats in the diabetic (D) group exhibited a significant reduction in body weight compared to the control (C) group (* *p* < 0.05). Treatment with *Glycyrrhiza glabra* (D+G group) significantly attenuated this weight loss compared to the untreated diabetic group (* *p* < 0.05). Subfigure (**B**): Shows the fasting blood glucose (FBG) levels across the experimental period. Diabetic rats (D group) had significantly elevated FBG levels compared to the control (* *p* < 0.05). The intervention with *Glycyrrhiza glabra* (D+G group) significantly reduced FBG levels compared to the diabetic group (* *p* < 0.05). The data are expressed as means ± standard deviation (n = 8–10). Statistical differences between groups are indicated by * *p* < 0.05.

**Figure 2 biology-14-00307-f002:**
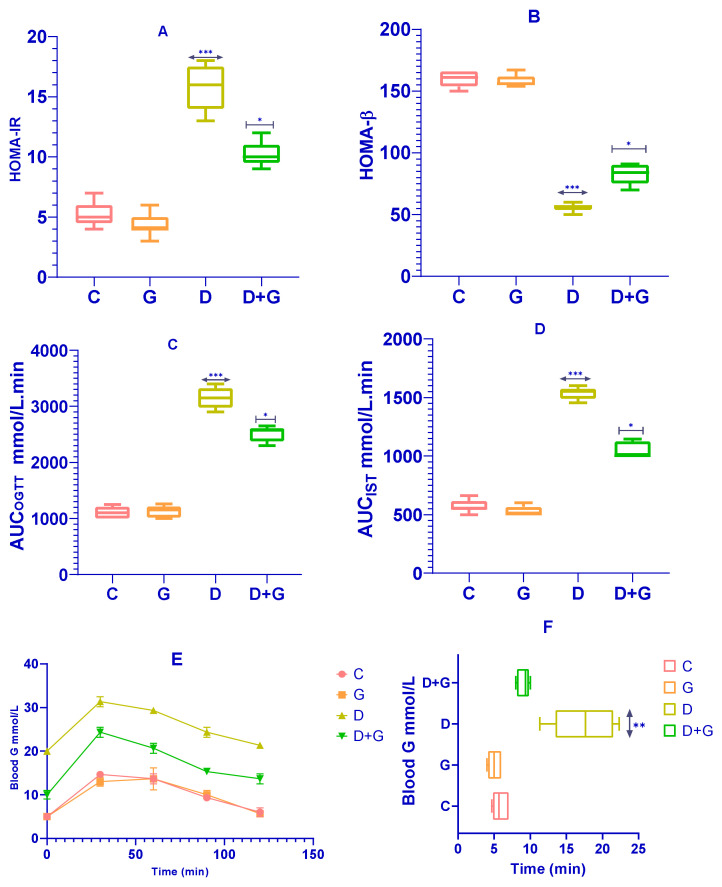
Subfigures (**A**–**F**) illustrate the impact of *Glycyrrhiza glabra* on glucose tolerance and insulin sensitivity in diabetic and control rats. Subfigure (**A**): Homeostatic Model Assessment for Insulin Resistance (HOMA-IR). Subfigure (**B**): Homeostatic Model Assessment for Beta Cell Function (HOMA-β). Subfigures (**C**,**D**): Area under the curve (AUC) for the Oral Glucose Tolerance Test (OGTT) and Insulin Sensitivity Test (IST), respectively. Subfigures (**E**,**F**): OGTT and IST results in week 13. The data are expressed as mean ± standard deviation (n = 8–10). Statistical significance is indicated as * *p* < 0.05, ** *p* < 0.001, *** *p* < 0.0001 versus the D group, and *p* < 0.05, *p* < 0.01 versus the G group.

**Figure 3 biology-14-00307-f003:**
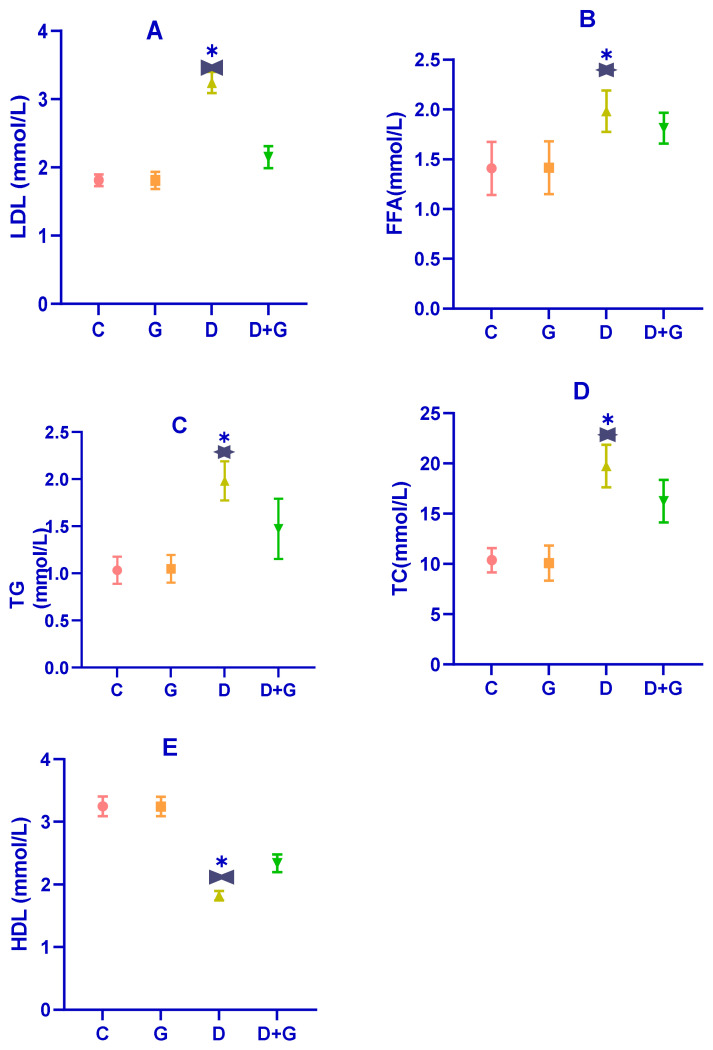
(**A**–**E**): Depicts the effects of an oral water extract of *Glycyrrhiza glabra* (G) on serum lipid levels in diabetic rats. The subplots illustrate the levels of (**A**) low-density lipoprotein (LDL), (**B**) free fatty acids (FFAs), (**C**) triglycerides (TG), (**D**) total cholesterol (TC), and (**E**) high-density lipoprotein (HDL). The data are expressed as mean ± standard deviation (n = 8–10). Statistical significance is indicated as: * *p* < 0.01 versus the D group, and * *p* < 0.05, versus the G group.

**Figure 4 biology-14-00307-f004:**
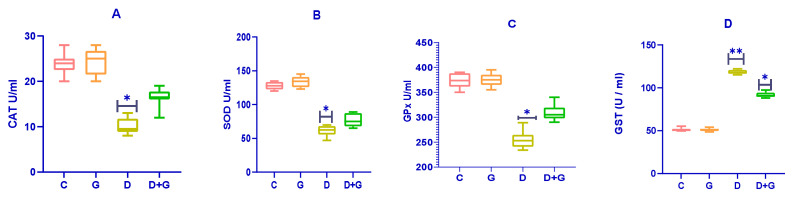
Subplots (**A**–**D**) demonstrate the effects of an oral water extract of *Glycyrrhiza glabra* (G) on antioxidant enzyme activities in diabetic rats. The subplots represent (**A**) catalase (CAT), (**B**) superoxide dismutase (SOD), (**C**) glutathione peroxidase (GPx), and (**D**) Glutathione-S-Transferase (GST). The activities are expressed as U/g tissue, with data presented as means ± standard deviation (n = 8–10). Statistical significance is denoted as *p* < 0.05, * *p* < 0.01 versus the D group, and ** *p* < 0.01 versus the G group.

**Figure 5 biology-14-00307-f005:**
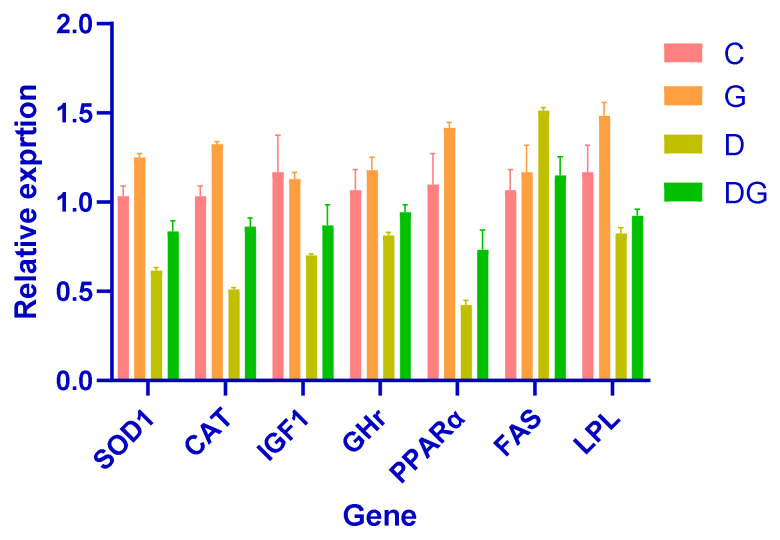
Illustrates the relative gene expression levels of SOD1, CAT, IGF1, GHr, PPARα, FAS, and LPL across the experimental groups (C: control; G: *Glycyrrhiza glabra*-treated; D: diabetic; DG: diabetic + *Glycyrrhiza glabra*-treated). The DG group showed significant upregulation of key genes, emphasizing the protective and restorative effects of *Glycyrrhiza glabra*.

**Figure 6 biology-14-00307-f006:**
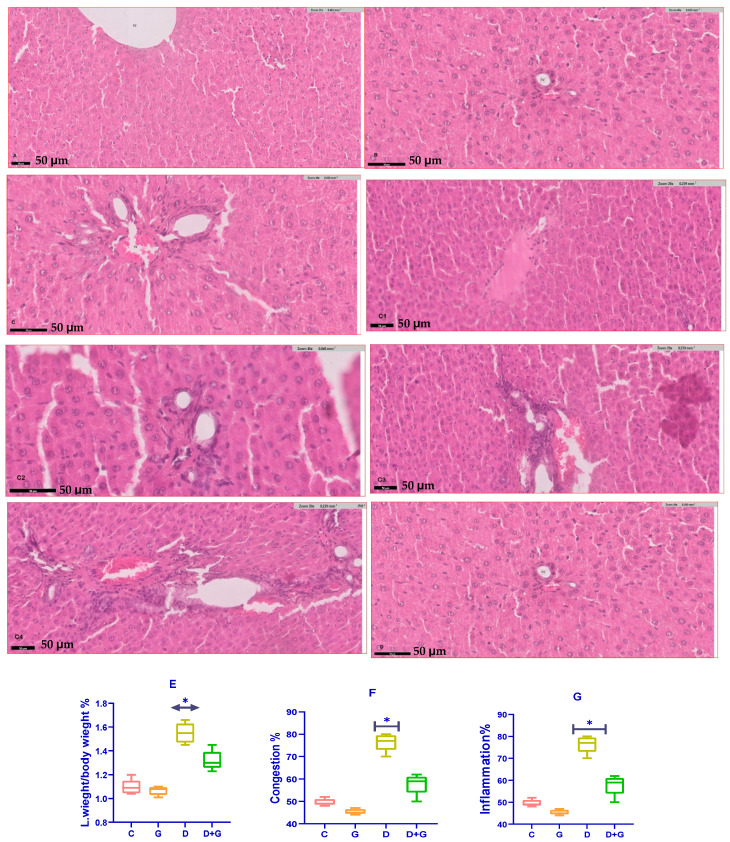
(**A**–**G**): Hepatoprotective effects of *Glycyrrhiza glabra* against liver tissue damage in diabetic rats. (**A**) Control group (**C**): Liver sections show normal histological architecture with intact hepatocytes, minimal fat deposition, and no signs of inflammation or fibrosis. (**B**) *Glycyrrhiza glabra*-treated group (**G**): Liver tissues exhibit preserved histological features similar to the control group, indicating no adverse effects of *Glycyrrhiza glabra* on liver morphology in healthy rats. (**C**–**C4**) Diabetic group (**D**): Liver sections reveal severe pathological changes, including significant fat deposition (steatosis), inflammatory cell infiltration, and fibrosis, indicating extensive liver damage due to diabetes and oxidative stress. (**D**) Diabetic + *Glycyrrhiza glabra*-treated group (DG): Marked improvement in liver histology is observed compared to the D group, with significantly reduced fat deposition and inflammatory infiltration and partial restoration of liver architecture, suggesting the protective effects of *Glycyrrhiza glabra*. (**E**) Quantification of fat deposition: A bar graph showing significantly higher fat deposition in the D group compared to the control (*p* < 0.001), while the DG group shows a substantial reduction in fat accumulation (*p* < 0.05). (**F**) Quantification of inflammatory cells: A bar graph showing a significantly higher inflammatory cell count in the D group compared to the control (*p* < 0.001), whereas the DG group exhibits a marked reduction in inflammation (*p* < 0.05). (**G**) Quantification of fibrosis: A fibrosis index graph highlighting significantly increased fibrosis in the D group compared to the control (*p* < 0.001), while the DG group shows a significant reduction in fibrosis levels (*p* < 0.05), indicating the antifibrotic effects of *Glycyrrhiza glabra*. Note: The symbol (*) indicates statistical significance between the compared groups (*p* < 0.05).

**Figure 7 biology-14-00307-f007:**
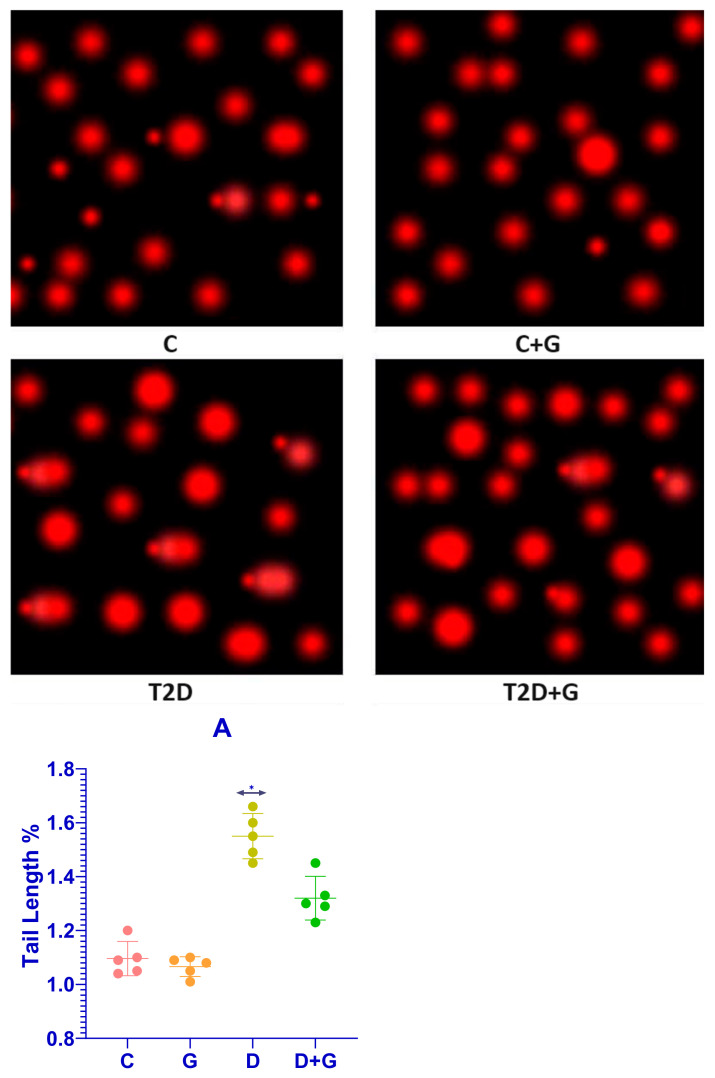
(**C**, **C+G**, **D**, **D+G**, **Panel A**): (**C**) Control group: Liver cells from the control group exhibited minimal DNA damage, with short apoptotic tails and an intact DNA structure. (**C+G**) Control + *Glycyrrhiza glabra* group: Similar to the control group, C+G cells demonstrated short apoptotic tails, indicating no adverse effects of *Glycyrrhiza glabra* on healthy liver cells. (**D**) Diabetic group: Liver cells in the diabetic group showed significantly longer apoptotic tails, reflecting extensive DNA damage caused by diabetes and oxidative stress. (**D+G**) Diabetic + *Glycyrrhiza glabra* group: The D+G group exhibited markedly shorter apoptotic tails compared to the D group, highlighting the protective effect of *Glycyrrhiza glabra* in reducing DNA strand breaks. (**A**) Dose–response curve: Illustrates the dose-dependent increase in apoptotic tail length across groups, with the D+G group displaying consistently shorter tail lengths at all radiation doses (2, 8, 30, and 60 Gy) compared to the D group. Note: The symbol (*) indicates statistical significance between the compared groups (*p* < 0.05).

## Data Availability

All datasets generated or analyzed during this study are included in the manuscript.

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
