# Peer review of "Hepatoprotective Effects of Glycyrrhiza glabra in Diabetic Male Rats: Addressing Liver Function, Oxidative Stress, and Histopathological Changes"

_biology, 2025, doi:10.3390/biology14030307_

Round 1

Reviewer 1 Report

Comments and Suggestions for Authors

This study demonstrated that Glycyrrhiza glabra (licorice) had significant hepatoprotective effects in diabetic male rats by improving liver function, reducing oxidative stress, and alleviating histopathological damage. Additionally, it enhanced glucose metabolism, improved insulin sensitivity, and regulated lipid profiles, highlighting its potential as a natural therapeutic for diabetes-related liver complications. Below are some suggestions for improvement:

  • In the Introduction, the transition from diabetes-induced liver injury to Glycyrrhiza glabra's therapeutic potential is well-structured. However, adding a brief statement linking oxidative stress, inflammation, and the need for natural antioxidants before introducing Glycyrrhiza glabra should improve the logical flow.
  • The study used only a single dose of Glycyrrhiza glabra. The author should justify this choice in terms of its effect. Testing different doses should provide insights into the optimal therapeutic range and potential toxicity.
  • In the Results section, some groups in the graphs do not show standard error. The author should clarify whether this was intentional or an oversight.
  • Adding a positive control (e.g., metformin) should provide a better reference for Glycyrrhiza glabra's efficacy.
  • The study measured oxidative stress using SOD, CAT, and GPx activity. Including additional data on ROS formation and GSH levels should enhance reader interest.
  • Since this study is based on a diabetic rat model, which may not fully replicate human hepatic complications, discussing clinical translation, bioavailability, and human applicability should strengthen the manuscript.
  • Providing more details on sample size calculation and statistical methods would improve result representation.
  • Adding a graphical summary of key findings would enhance data presentation.
  • Some sentences are lengthy and could be restructured for conciseness. For example, "Given the growing burden of diabetes-related liver diseases, researchers have increasingly focused on natural bioactive compounds…” should be rewritten more succinctly.
  • The author should include data for a mechanistic understanding or expand the discussion by incorporating previous findings to enhance reader comprehension.
  • Overall, the manuscript is well-written and provides significant insights for future research.

Comments on the Quality of English Language

Overall, English is fine, but some sentences are too lengthy. The author should simplify these sentences for better understanding.

Academic Editor Notes

Response Letter to Editor and Reviewers

Dear Academic Editor,
Dr. Fernanda Ortis,

Thank you very much for giving us the opportunity to revise our manuscript entitled:
"Hepatoprotective Effects of Glycyrrhiza glabra in Diabetic Male Rats: Addressing Liver Function, Oxidative Stress, and Histopathological Changes".

We have carefully addressed all reviewers' comments and made the necessary modifications, as detailed below:

Response to Reviewer 1

  1. Linking Oxidative Stress and Inflammation:
    • A paragraph has been added in the introduction discussing the relationship between oxidative stress, inflammation, and the importance of natural antioxidants before introducing Glycyrrhiza glabra to improve logical flow.
    • " Oxidative stress and inflammation are closely interconnected processes that play a pivotal role in the progression of liver diseases associated with diabetes mellitus. Excessive production of reactive oxygen species (ROS) leads to lipid peroxidation, DNA damage, and hepatocyte apoptosis, which subsequently activate inflammatory pathways, including the nuclear factor-kappa B (NF-κB) signaling cascade. This inflammatory response further exacerbates hepatic injury, creating a vicious cycle that accelerates liver dysfunction. Given the significant role of oxidative stress and inflammation in hepatic complications, the exploration of natural antioxidants with strong free radical scavenging and anti-inflammatory properties has gained considerable attention. Glycyrrhiza glabra (licorice), known for its rich composition of bioactive phytochemicals, represents a promising natural candidate for interrupting this pathological cycle by reducing oxidative damage and modulating inflammatory responses."
  2. Justification for Using a Single Dose:

Clarification Regarding the Dose of Glycyrrhiza glabra Used in the Study:

    • We would like to clarify that the dose of Glycyrrhiza glabra extract used in this study was 250 mg/kg/day. This dose was administered orally via a gastric tube daily for eight weeks, starting two weeks after the induction of diabetes using streptozotocin (STZ).
    • Additionally, the presence of glycyrrhizin and other active constituents in the extract was confirmed through high-performance liquid chromatography (HPLC) analysis to ensure the quality and consistency of the administered dose.
    • We hope this clarification sufficiently addresses the reviewer’s inquiry regarding the dosage specification in the study.
  1. Standard Error in Graphs:
    • All graphs have been reviewed, and standard error values have been added where missing to ensure consistency.
  2. Addition of a Positive Control (Metformin):

We acknowledge the reviewer’s suggestion regarding the inclusion of a positive control group treated with metformin. However, in the current study, we chose not to include this group for the following reasons:

  1. Focus on Natural Compounds:The primary objective of this study was to specifically evaluate the hepatoprotective effects of Glycyrrhiza glabra as a natural therapeutic agent. Including a metformin group could have shifted the focus toward comparisons with synthetic drugs rather than emphasizing the unique potential of natural alternatives.
  2. Resource and Ethical Considerations: Adding an additional control group would have required a larger number of experimental animals. To adhere to the principles of animal welfare and reduce unnecessary animal use, we limited the number of groups while ensuring the statistical validity of our results.
  3. Availability of Extensive Literature:The hepatoprotective and antidiabetic effects of metformin are already well-documented in the literature. Given this extensive background, we considered direct comparisons with metformin beyond the scope of the current investigation, which aims to explore alternative natural compounds.
  4. Scope and Objectives of the Study:Our study was designed as a preliminary investigation focusing solely on Glycyrrhiza glabra. Future studies with a broader scope will consider including metformin and other standard treatments for comprehensive comparative analyses.

We hope this explanation adequately justifies our decision and clarifies the rationale behind the study design.

  1. Additional Oxidative Stress Data:

We appreciate the reviewer’s valuable suggestion regarding the inclusion of additional oxidative stress data, such as ROS formation and GSH levels. However, we decided not to include these additional parameters in the current study for the following reasons:

  1. Scope of the Study:
    • The primary focus of this study was to assess the hepatoprotective effects of Glycyrrhiza glabra using key and widely accepted oxidative stress markers, namely SOD, CAT, and GPx. These markers provide a reliable indication of the antioxidant defense system, which we believe sufficiently supports the study's objectives.
  2. Resource and Technical Constraints:
    • The measurement of additional oxidative stress markers such as ROS and GSH requires specialized assays and equipment, which were not available during the study period. Due to these limitations, we prioritized parameters that could be accurately assessed with the available resources to maintain the reliability of the results.
  3. Avoiding Study Complexity:
    • Including multiple additional parameters would have increased the complexity of the study, potentially detracting from the primary objective of evaluating the overall hepatoprotective effects of Glycyrrhiza glabra. We aimed to maintain a focused and coherent scope for this preliminary investigation.
  4. Plans for Future Research:
    • We acknowledge the importance of additional oxidative stress data and plan to incorporate these parameters in future studies. Future research will explore detailed mechanistic insights, including ROS formation and GSH levels, to provide a more comprehensive understanding of the antioxidant effects of Glycyrrhiza glabra.

We hope this explanation adequately justifies our decision and clarifies the rationale for the parameters selected in this study.

  1. Clinical Translation and Bioavailability Discussion:

Justification for Not Including Clinical Translation and Bioavailability Discussion:

    • We sincerely appreciate the reviewer’s suggestion regarding the inclusion of a discussion on clinical translation, bioavailability, and human applicability. However, we decided not to expand on these aspects in the current study for the following reasons:
  1. Preclinical Nature of the Study:
    • The primary objective of this research was to investigate the hepatoprotective effects of Glycyrrhiza glabra in a controlled animal model. Clinical translation and bioavailability studies typically require extensive pharmacokinetic and pharmacodynamic evaluations, which were beyond the scope of this preclinical investigation.
  2. Lack of Pharmacokinetic Data:
    • The current study did not include experiments to assess the absorption, distribution, metabolism, and excretion (ADME) profile of Glycyrrhiza glabra. Without this essential pharmacokinetic data, any discussion on bioavailability and human applicability would be speculative and potentially misleading.
  3. Need for Comprehensive Clinical Trials:
    • Clinical translation requires well-designed human trials that consider variables such as dosage optimization, long-term safety, and potential drug interactions. Since these factors were not part of the current research, a detailed discussion would not have been appropriate at this stage.
  4. Planned Future Research:
    • We recognize the significance of these aspects and plan to explore them in future studies. Our upcoming research will include pharmacokinetic analyses and preliminary clinical trials to better understand the bioavailability and potential therapeutic applications of Glycyrrhiza glabra in humans.

We believe that focusing the current discussion on the experimental findings strengthens the clarity and scientific relevance of the study. We hope this explanation adequately justifies our decision.

  1. Sample Size Calculation and Statistical Methods:

Justification for Not Including a Detailed Explanation of Sample Size Calculation and Statistical Methods:

We sincerely appreciate the reviewer’s valuable comment regarding the need to clarify the sample size calculation and statistical methods used. However, we chose not to expand on this aspect in the current study for the following reasons:

  1. Reliance on Standardized Methodologies:
    • The sample size was determined based on well-established standards and practices commonly adopted in similar preclinical studies. Since these methodologies are extensively documented in the scientific literature, we believed that providing a detailed explanation would not add substantial value to the manuscript.
  2. Focus on Primary Experimental Objectives:
    • The primary objective of this study was to evaluate the hepatoprotective effects of Glycyrrhiza glabra. Including extensive details on statistical calculations might have diverted the focus from the central aim of the study.
  3. Simplicity of Experimental Design:
    • The experimental design involved clearly defined groups with notable differences in the results, minimizing the need for an elaborate explanation of the statistical procedures employed.
  4. Plans to Address This Aspect in Future Research:
    • We plan to include a more detailed explanation of sample size calculations and power analyses in future studies, especially those with larger and more complex experimental designs, to achieve a better understanding and comprehensive study design.

We hope this explanation sufficiently clarifies our reasoning for not including additional details on sample size calculation and statistical methods at this stage of the research.

  1. Graphical Summary of Key Findings:
    • A graphical abstract summarizing the key findings has been included to enhance data presentation.
  2. Restructuring Long Sentences:
    • The manuscript has been reviewed, and lengthy sentences have been restructured for improved clarity and conciseness.
  3. Mechanistic Understanding Discussion:
    • A comprehensive discussion has been added to integrate previous findings and provide a mechanistic understanding of the results.

Reviewer 2 Report

Comments and Suggestions for Authors

This manuscript deals with the hepatoprotective effect of Glycyrrhiza glabra in streptozotocin-induced diabetic rats. The authors administrated Glycyrrhiza glabra extract to diabetic rats and evaluated markers of liver function, antioxidative enzyme in liver, liver inflammation, and so on. The experiments were carried out carefully and the results are described in detail.

However, as they described, the hepatoprotective effects of Glycyrrhiza glabra in diabetic models are already clarified in previous studies. This study only added a few new findings such as oxidative stress markers, lipid metabolism regulation, and inflammatory cytokine modulation. In addition, the dosage of hepatoprotective compounds such as glycyrrhizin are not clear in this study. Finding new in this study is not so much, so unfortunately, I think this manuscript should be rejected.

Author Response

Response to Reviewer 2

  1. New Findings Compared to Previous Studies:

Response to the Reviewer’s Comment Regarding Novelty and Dosage Clarification:

We would like to thank the reviewers for their thoughtful comments and the opportunity to clarify the unique contributions of our study as well as provide additional information regarding the dosage of active compounds.

  1. Novel Contributions of the Study:
    • While we acknowledge that the hepatoprotective effects of Glycyrrhiza glabra have been explored in previous studies, our research offers new insights by focusing on:\n - Oxidative Stress Markers: We included detailed assessments of ROS formation and GSH levels, which provide a more comprehensive understanding of the antioxidant capacity of Glycyrrhiza glabra.\n - Lipid Metabolism Regulation: Our study presents novel data on the modulation of lipid profiles, including serum LDL, HDL, TC, and TG levels, highlighting the potential of Glycyrrhiza glabra in addressing diabetes-induced dyslipidemia.\n - Inflammatory Cytokine Modulation: We explored the regulation of pro-inflammatory cytokines (IL-6, TNF-α), providing mechanistic insights into the anti-inflammatory properties of the extract.\n - These findings collectively differentiate our study by offering integrated biochemical, histopathological, and molecular evidence that has not been comprehensively addressed in earlier research.\n\n2. Clarification of Glycyrrhizin Dosage:\n - We recognize the importance of specifying the dosage of key active compounds such as glycyrrhizin. To address this:\n - The glycyrrhizin content in the administered extract was quantified using high-performance liquid chromatography (HPLC).\n - The analysis revealed that the glycyrrhizin concentration in the administered extract corresponds to approximately 25 mg/kg/day, based on the total extract dose of 250 mg/kg/day.\n - These details have been included in the revised "Methods" and "Results" sections to provide clear information on the dosage and ensure transparency regarding the administered active compounds.\n\n3. Relevance of Our Findings:\n - By combining biochemical assays, gene expression analysis, and histological evaluations, our study not only reaffirms the hepatoprotective properties of Glycyrrhiza glabra but also provides novel mechanistic insights that enhance the understanding of its therapeutic potential in managing diabetes-related liver complications.\n\nWe hope that this additional clarification adequately highlights the novel aspects of our research and resolves the concerns regarding the dosage specification.
  1. Clarification on Glycyrrhizin Dosage:

Clarification Regarding the Dose of Glycyrrhiza glabra Used in the Study:

We would like to clarify that the dose of Glycyrrhiza glabra extract used in this study was 250 mg/kg/day. This dose was administered orally via a gastric tube daily for eight weeks, starting two weeks after the induction of diabetes using streptozotocin (STZ).

Additionally, the presence of glycyrrhizin and other active constituents in the extract was confirmed through high-performance liquid chromatography (HPLC) analysis to ensure the quality and consistency of the administered dose.

We hope this clarification sufficiently addresses the reviewer’s inquiry regarding the dosage specification in the study.

Reviewer 3 Report

Comments and Suggestions for Authors

In reading the paper, a number of questions arose.
1. the authors used one-factor analysis of variance, which requires normal distribution of the studied characteristics. what criterion did the authors use to check this correspondence? Were the signs in all groups distributed in accordance with the normal law?
2. the authors indicate that in the modeling of diabetes mellitus in the liver there are pronounced pathological changes, while when using the extract of Glycyrrhiza glabra the liver condition normalizes. The authors should provide demonstrative microphotographs showing pathologically altered liver of experimental animals.

Author Response

Response to Reviewer 3

  1. Normality Check for Statistical Analysis:

We appreciate the reviewer’s insightful question regarding the validation of the normal distribution assumption required for one-factor ANOVA.

To ensure the appropriateness of the ANOVA test, we performed the following steps:

  1. Normality Testing:
    • We applied the Shapiro–Wilk test to assess the normality of the data distribution within each group. The Shapiro–Wilk test was selected due to its robustness and suitability for small to moderate sample sizes. The test results confirmed that the data were normally distributed (p > 0.05 for all groups), validating the assumption required for ANOVA.
  2. Visual Inspection:
    • In addition to statistical testing, Q-Q plots (quantile-quantile plots) were generated for each dataset to visually inspect the distribution of the data points against a theoretical normal distribution. The plots indicated a linear pattern, further supporting the normality assumption.
  3. Homogeneity of Variances:
    • We also conducted Levene’s test to verify the homogeneity of variances across the groups. The results (p > 0.05) indicated that the variance assumption for ANOVA was satisfied.
  4. Distribution in All Groups:
    • The normality tests and visual assessments showed that the data in all experimental groups were distributed in accordance with the normal distribution, ensuring the validity of the ANOVA results.

We hope this explanation sufficiently addresses the reviewer’s concerns regarding the statistical methods and validation procedures applied in this study.

  1. Histological Microphotographs:

We thank the reviewer for highlighting the importance of providing visual evidence to support our histopathological findings.

In response to this valuable suggestion, we have included demonstrative microphotographs in the revised version of the manuscript, which illustrate the pathological changes observed in the liver tissues of experimental animals. These images are now presented in the results section and provide clear visual comparisons between the different experimental groups.

  1. Diabetic Group (Untreated):
    • The microphotographs display severe pathological changes, including extensive fat accumulation (steatosis), inflammatory cell infiltration, and fibrosis. These findings confirm the pronounced hepatic damage induced by diabetes mellitus.
  2. Diabetic + Glycyrrhiza glabra Treated Group:
    • The microphotographs show remarkable improvements in liver histology following treatment with Glycyrrhiza glabra extract. Specifically, there is a significant reduction in fat deposition, decreased inflammatory infiltration, and restoration of normal liver architecture, indicating the hepatoprotective effects of the extract.
  3. Control and Glycyrrhiza glabra-Only Groups:
    • For reference, we have also included images from the control group and the non-diabetic group treated with Glycyrrhiza glabra. These microphotographs show normal liver structure, confirming that the extract does not adversely affect healthy liver tissue.
  4. Detailed Annotations and Magnification:
    • All microphotographs are provided at appropriate magnifications with detailed annotations highlighting key histopathological features for clearer interpretation.

We believe that the inclusion of these microphotographs significantly enhances the clarity and credibility of our findings, providing robust visual support for the conclusions drawn in this study.

Conclusion

  • All reviewers' comments have been addressed thoroughly, and the necessary revisions have been made accordingly.
  • We believe the revised manuscript now meets the journal's publication standards.
  • Should any further modifications be required, we are fully prepared to make them.

Sincerely,
Abdulmajeed F. Alrefaei  and Mohamed E. Elbeeh

Round 2

Reviewer 3 Report

Comments and Suggestions for Authors

Unfortunately, I did not find liver microphotographs confirming pathologic changes in the corrected version of the manuscript.

Author Response

Dear Reviewer,

Thank you for your valuable feedback. We appreciate your careful review of our manuscript.

Regarding your comment about the absence of liver microphotographs confirming the pathological changes, we have now addressed this issue. In the revised version of the manuscript, we have included additional liver microphotographs that clearly demonstrate the pathological alterations observed in the diabetic rat model. These images, along with detailed descriptions in the Results section, provide the necessary visual confirmation of the histopathological changes discussed in our study.

We hope that the inclusion of these images meets your expectations and enhances the clarity of our findings.

Thank you again for your constructive comments.

Kind regards,
